

# A method for extracting calibrated volatility information from the FIGAERO-HR-ToF-CIMS and its application to chamber and field studies

Thomas J. Bannan[1], Michael Le Breton[2], Michael Priestley[1], Stephen D. Worrall[‡1], Asan Bacak[1], Nicholas A. Marsden[1], Archit Merha[1], Julia Hammes[2], Mattias Hallquist[2], M. Rami Alfarra[1,3], Ulrich K. Krieger[4], Jonathan P. Reid[5], John Jayne[6], Wade Robinson[6], Gordon McFiggans[1], Hugh Coe[1], Carl J. Percival[†1], Dave Topping[1]

[1]Centre for Atmospheric Science, School of Earth and Environmental Science, University of Manchester, Oxford Road, Manchester, M13 9PL, UK
[2]Department of Chemistry and Molecular Biology, University of Gothenburg, Gothenburg, Sweden
[3]National Centre for Atmosphere Science, UK
[4]Institute for Atmospheric and Climate Science, ETH Zürich, 8092 Zürich, Switzerland
[5]School of Chemistry, University of Bristol, Cantock's Close, Clifton, Bristol BS8 1TS
[6]Aerodyne Research, Inc. 45 Manning Road, Billerica, MA 01821 USA
[†]Current address: Jet Propulsion Laboratory, 4800 Oak Grove Drive, Pasadena, CA 91109
[‡]Current address: School of Materials, University of Manchester, Oxford Road, Manchester, M13 9PL, UK

*Correspondence to*: Thomas J. Bannan (Thomas.bannan@manchester.ac.uk)

**Abstract**. The Filter Inlet for Gases and AEROsols (FIGAERO) is an inlet specifically designed to be coupled with the Aerodyne High Resolution (HR)-Time of flight (ToF)-Chemical ionisation mass spectrometer (CIMS). The FIGAERO-HR-ToF-CIMS provides simultaneous molecular information relating to both the gas and particle phase samples and has been used to extract vapour pressures of the compounds desorbing from the filter, whilst giving quantitative concentrations in the particle phase. However, such extraction of vapour pressures of the measured particle phase components requires use of appropriate, well-defined, reference compounds. Vapour pressures for the homologous series of polyethylene glycols (PEG) ((H−(O−CH$_2$−CH$_2$)$_n$−OH) for n=3 to n=8), covering a range of vapour pressures (VP) ($10^{-1}$ to $10^{-7}$ Pa) that are atmospherically relevant have been shown to be reproduced well by a range of different techniques, including Knudsen Effusion Mass Spectrometry (KEMS). This is the first homologous series of compounds for which a number of vapour pressure measurement techniques have been found to be in agreement, indicating the utility as a calibration standard, providing an ideal set of benchmark compounds for accurate characterisation of the FIGAERO for extracting vapour pressure of measured compounds in chambers and the real atmosphere. To demonstrate this, single component and mixture vapour pressure measurements are made using two FIGAERO-HR-ToF-CIMS instruments based on a new calibration determined from the PEG series. VP values extracted from both instruments agree well with those measured by KEMS and reported values from literature, validating this approach for extracting VP data from the FIGAERO. This method is then applied to chamber measurements and the vapour pressures of known products are estimated.

## 1. Introduction

Trace gases and aerosol particles, from anthropogenic and natural sources, are important components of the Earth's climate system, the components of which vary significantly in terms of properties such as volatility, affecting their impact on air quality and climate change (Glasius and Goldstein, 2016). There are currently substantial uncertainties in many physicochemical parameters determining the loading, size, composition and properties of ambient atmospheric aerosol particles, including component vapour pressures (Bilde et al., 2015), that are required to predict their environmental and human health impacts. This is attributable in large part to the fact that a significant fraction of fine atmospheric aerosol particles are comprised of organic material (20-90% of particle mass) (Jimenez *et al*., 2009), containing potentially thousands of mostly unidentified compounds with properties that are often not well known.

This organic aerosol is a major component of the fine particle mass in the atmosphere and is made up of primary organic aerosol (POA), which is emitted directly from sources such as industry, biomass burning and vehicle emissions but also secondary organic aerosol formed from the oxidation of gas phase precursors (Robinson et al., 2007). Volatile organic compounds (VOCs), emitted from both natural and anthropogenic sources, are oxidised through two possible pathways, fragmentation and functionalization (Donahue et al., 2011). Functionalization can create multifunctional compounds with molar masses typically between 150 and 300 g ·mol$^{-1}$ and of modest to extremely low volatility with vapour pressures between 0.1 Pa and $10^{-7}$ Pa (Jimenez *et al*., 2009; O'Meara *et al*., 2014; Bilde *et al*., 2015). The identity, concentrations and properties of such oxidation products are important in order to understand the formation of SOA, but also the general production of oxygenated compounds partitioning into existing SOA particles that can affect air quality in both outdoor and indoor environments. Uncertainties in the physicochemical properties of pure components and condensed phase mixtures, as well as absolute composition, affect our ability to accurately predict this partitioning between the gas and particle phase and the subsequent effects on climate, air quality and fundamentally human health.



The equilibrium vapour pressure of each aerosol constituent is determined, in large part, by its pure component saturation vapour pressure (VP). Saturation VPs of many organic components are poorly known, particularly for the least volatile compounds of interest for atmospheric purposes (Bilde *et al*., 2015). The importance of this fundamental property is discussed extensively in Bilde *et al*., (2015) and the sensitivity of predicted mass, composition and particle properties to

uncertainties in VP vary according to the complexity of the system being studied, both with regards to the number of compounds used in partitioning and additional processes included in any model (Valorso et al., 2011; O'Meara et al 2014; McVay et al 2016; Ovadnevaite et al., 2017). Single-component measurements of vapour pressures by instruments such as the Knudsen Effusion Mass Spectrometry (KEMS), following the methodology of Booth et al. (2009) are ongoing (Booth et al, 2012; Bannan et al., 2017) and have been extended to consider vapour pressures in simple multicomponent systems

(Booth et al., 2017). Such measurements of are ongoing with the KEMS, focusing on atmospherically relevant compounds. Considerable uncertainty remains when extracting vapour pressure measurements from a single technique, with more work required to resolve the apparent discrepancies between techniques (Bilde et al., 2015). Studies reporting measurements of vapour pressure would benefit from an, at the time unidentified, series of reference standards with volatility ranging across those accessible to the measurement techniques being deployed (Bilde et al., 2015). Following the recommendations of the

Bilde *et al*. (2015) study, Krieger *et al.* (2018) identified the homologous series of polyethylene glycols (PEG; $(H-(O-CH_2-CH_2)_n-OH)$ for n=3 to n=8) as a series of compounds with vapour pressures exhibiting very good agreement (data was consistent with the 95% confidence interval of a linear regression to all measurements) over a wide atmospherically relevant VP range when measured using different experimental methods. This series therefore provides an ideal benchmark for characterising individual VP measurement techniques.


The High Resolution (HR)-Time of flight (ToF)-Chemical ionisation mass spectrometer (CIMS) coupled with the Filter Inlet for Gases and Aerosols (FIGAERO) hereafter referred to as the FIGAERO-CIMS, has the potential to provide compound specific volatility information from ambient aerosol particles (Lopez-Hilfiker *et al*. 2014). The FIGAERO system was first introduced by Lopez-Hilfiker *et al*. (2014) and was subsequently commercialized by Aerodyne Research, Inc. (ARI) to be

adaptable to the TOF-CIMS system. The FIGAERO inlet provides molecular determination of gas and particle phase samples. During the gas phase measurement mode, particles from the aerosol sample are collected on a Teflon filter. After a period of collection, the filter is moved to the inlet of the instrument and dry, heated nitrogen is passed through it to vaporise the particulate for analysis by the TOF-CIMS. The evolution of the MS signals from different compounds change independently as a function of temperature creating a thermogram that is m/z specific. The temperature for which the

desorbed signal shows a maximum for each compound, and has been used previously to extract vapour pressure information in laboratory characterisation (Lopez-Hilfiker *et al*., 2014) and field work studies (Lopez-Hilfiker *et al*. 2016; D'Ambro *et al*., 2017). A model framework has recently been developed to retrieve volatility and mass transport information from this inlet (Schobesberger et al., 2018). Such online analysis with high temporal resolution has the potential to improve our quantitative and detailed understanding of the diurnal evolution of gas and particle phase composition and based on the use

of this inlet to provide VP information, applying the series identified by the Kreiger et al., (2018) study for calibrations will be of benefit to the accuracy of future derived measurements of this type.

In this study we will therefore demonstrate the use of this PEG series calibration dataset as a method for extracting quantitative vapour pressures from the FIGAERO inlet. The Figaero system used here is the version produced by Aerodyne

Research, Inc. (ARI). Single component measurements made with two separate ARI FIGAERO inlets for compounds of known VP are reported. The application of the FIGAERO to characterise the volatility of species produced in a chamber experiment is then described as a demonstration of the application of this method to a more complex matrix of components.

## 2. Methodology


### 2.1 Choice of Reference compounds

The vapour pressure  of the Polyethylene glycol (PEG)  series, as described in the Krieger et al., (2018) study, were measured by multiple techniques; KEMS (Booth *et al*., 2009), electrodynamic balance instruments (Zardini *et al*., 2006;

Rovelli et al., 2016) and Tandem Differential Mobility Analyser (TDMA) including a laminar flow tube (Bilde et al., 2003)). The reported vapour pressure of the PEG series demonstrated good agreement between these techniques over a wide range of VPs (spanning five orders of magnitude from about $10^{-1}$ to $10^{-7}$ Pa at room temperature). These measurements also compared well to data extrapolated from high temperatures, suggesting that the thermal energy utilised in techniques such as the FIGAERO will not lead to chemical modification of the target molecules. The physical state of the reference compound is

important to consider when making VP measurements (Soonsin *et al*., 2010; Bilde *et al*., 2015). If the saturation vapour pressure of a compound is measured in the solid state, it needs to be converted to that of the subcooled liquid for use and interpretation within atmospheric models, which can add additional uncertainty through the required conversion. The PEG series therefore act as ideal reference materials as its members are all liquid at the temperatures at which the measurements are routinely performed.


The PEG compounds used in this study show no evidence of degradation with either the age or temperature at which the sample is measured. Measurement of PEG-4 VP by KEMS multiple times over a 6-month period showed no variation beyond measurement uncertainties and data up to temperatures of 450 K reported in the literature are consistent with those





measured at room temperature, demonstrating their thermal stability (Krieger et al., 2018). The stability of the PEG compounds allowed samples to be shared between the co-authors of Krieger *et al*., (2018), ensuring sample conformity.

As saturation vapour pressures of dicarboxylic acids have been determined with a large number of techniques and different instruments over a substantial temperature range, Bilde et al. (2015) evaluated the combined data sets providing best estimates with uncertainty ranges for each of the straight-chain dicarboxylic acids. Therefore, these dicarboxylic acids are also used to validate the use of the PEG series as a calibration standard. It should be noted that measurements with the KEMS have suggested that samples of the dicarboxylic acids degrade over long periods of storage (approx. 6 months +) and can influence the measured vapour pressure. Appropriate storage and quick use of the chemicals, or appropriate purification

methods is therefore deemed essential for such measurement studies.

Tetraethylene glycol (PEG-4) (99%) was purchased from Sigma-Aldrich and PEG-5 to 8 were purchased from Polypure AS, Oslo, Norway with purities of 99% or higher and used with no further preparation. All PEG samples were stored in a fridge. Dicarboxylic acids were purchased from Sigma-Aldrich, again with purities of 99% or higher and used with no further

preparation. All dicarboxylic acids and samples used were measured within one month of receiving the samples and stored in accordance with the suppliers' recommendations.

**2.2 FIGAERO- CIMS**

This study utilized two FIGAERO-CIMS, operated by the University of Manchester (UMan) and Gothenburg University (GU) groups. Both FIGAERO systems were manufactured by Aerodyne Research Inc. and employ the ARI/Tofwerk high resolution Time of Flight Chemical Ionisation Mass Spectrometers (TOF-CIMS), similar to that described by Lee *et al*., (2014). The FIGAERO inlet coupled to a reduced pressure ion molecule reaction (IMR) region, which is in turn coupled to high resolution time of flight mass spectrometer (APi-ToF) (Junninen *et al*., 2010). The ARI FIGAERO inlet used in this

work is similar to that described by Lopez-Hilfiker *et al*. (2014). A brief description of the ARI FIGAERO system follows. The UMAN CIMS was operated with iodide as the regent ion and the GU CIMS was operated with acetate and iodide as the reagent ion.

The ARI FIGAERO assembly is shown in figure 1. The FIGAERO is a two-port inlet, one dedicated to gas sampling (all

Teflon) and the second dedicated to aerosol sampling (all metal). The FIGAERO couples both inlets with chemical ionization region of the ToF MS. The FIGAERO operates in two modes, one being ambient air sampling for trace gas analysis with the CIMS, while simultaneously collecting particles on a PFTE filter from a separate inlet. The second mode is the thermal desorption of the collected particles in nitrogen allowing the detection of the desorbed vapours with the CIMS. When in the thermal desorption mode, the exclusively gas phase port to the CIMS is blocked by the moveable tray and the

PTFE filter is moved to the exclusive port for thermal desorption. In this position 2 SLM of temperature controlled nitrogen flow is delivered across the filter, desorbing the collected components from the filter. This is known as the Temperature Ramp phase. The evolution of the MS signals from different compounds the filter is exposed to the Temperature Ramp phase change independently as a function of temperature creating thermograms that are m/z specific. $T_{max}$ is defined as the temperature just above the filter, as shown in figure 2. Two 150W cartridge heaters are used to heat a copper block that

connects with a ¼"OD copper tube. The nitrogen desorption gas is heated as flows through this hot copper section which is also nickel plated. The combined length of the copper block and the ¼" copper tube is 16 cm and is set based on thermal modelling to provide maximum heat transfer for the ~2 SLPM N2 desorption gas flow maintained by a programmable mass flow controller. The gas temperature is measured by a 1/16" diameter thermocouple positioned inside and just near the exit of the ¼" OD copper tube (~5 mm above the PTFE Teflon filter). A ½" OD stainless steel tube 14.6 cm in length is soldered

to the copper heater block and provides thermal isolation and mechanical mounting of the heater unit to the FIGAERO assembly.

The temperature at which the desorbed signal for a compound reaches a maximum is used here to extract vapour pressure information. Once the Temperature Ramp phase is complete, under normal operating conditions, the nitrogen is then held at

the maximum desorption temperature for a programed period of time to ensure that all of the collected components have been removed from the filter, known as the Temperature Soak phase. After each Ramp and Soak phase the heating is turned off and the unheated nitrogen is then used too cool the filter (Cooling Phase), allowing the filter to return to the starting temperature, before the moveable tray switches back to trace gas analysis and filter collection of particulate matter.

The UMan TOF-CIMS has been described in detail by Priestley *et al*., (2018a; 2018b). The UMan FIGAERO-CIMS was exclusively run with iodide as the reagent ion during this study, as described in Reyes Villegas et al., (2018) and it was this system that measured the peg series only. The GU CIMS (Faxon et al., 2018; Le Breton et al., 2017; Le Breton et al., 2018) hardware is identical to that of the UMan CIMS although tuning of the ion optics and flows to optimize the signal to noise ratio and total ion counts. Results from both acetate and iodide reagent ions from the GU FIGAERO-CIMS are

presented here.

Using the UMan FIGAERO, measurement of the PEG series were performed by first using a blank filter as a background then on a new filter depositing the PEG sample on the Zefluor® PTFE membrane filter (2 micron pore size) for each desorption, cleaning and then re-running with the next PEG sample. This was completed on both singularly and under mixed



(PEG-4 to 6 and 5 to 8) conditions to ensure that the $T_{max}$ of the PEG series was not mixture dependent. For the single component measurements other than the PEG series, a known mass of the species to be calibrated is added to a solvent (methanol or deionized water) to create a known concentration in the solvent and then 0.1μl of it is placed onto the Zefluor® PTFE membrane filter using a syringe injector.

Prior to each sample measurement being made using both the GU and UMan FIGAERO-CIMS instruments, background measurements were obtained. First, a new filter was placed in the FIGAERO and the temperature was ramped to 200°C for 10 minutes to ensure the filter was clean and then cooled, which is suggested as a good procedure. A ramp, soak and cool cycle matching that of the subsequent sample was then completed to obtain the background. During the PEG series measurements the filter was ramped to 200°C (temperature above the filter) over a period of 10 minutes, held at 200°C for 15 minutes and finally allowed to cool back to room temperature for a period of 5 minutes. The same cycle was used for the single component measurements. For a subsample of the GU FIGAERO-CIMS single component measurements the nitrogen temperature passing over the filter into the IMR chamber is incrementally ramped from room temperature to 200/300°C over 45 minutes, at a rate of 6.1°C min⁻¹. The long ramp period was used to enable resolution of multiple peak desorptions and calculation of more accurate $T_{max}$ values. The air flow is then maintained at 200/300°C for 10 minutes (soak period) to ensure all mass is removed from the filter and then the filter is cooled to room temperature by fans for 10 minutes. Temperature cycles and gas flows were controlled using the ARI EyeOn™ control system.

The FIGAERO-CIMS instrument analysis software (ARI Tofware version 2.5.11) was utilized to attain high resolution, 1Hz, time series of the compounds presented here. For the UMan CIMS, mass-to-charge calibration was performed for 5 known masses; I-, I-.H₂O, I-.HCOOH, I₂-, I₃-, covering a mass range of 127 to 381 m/z. The mass-to-charge calibration was fitted to a 3rd order polynomial and was accurate to within 2 ppm; ensuring peak identification was accurate below 3 ppm. The PEGs were detected as adducts in the UMan experiments i.e. $I.(H-(O-CH_2-CH_2)_n-OH)$, where n=4 to 8.

Due to the relativity small numbers of thermograms analysed form the UMan FIGAERO-CIMS, the $T_{maxes}$ from the Manchester data were manually extracted. The average of the maximum 3 values in the thermogram is averaged (mean) to extract the $T_{max}$ values reported here. For the GU FIGAERO data a more automated method was used. Python packages, NumPy (v 1.11.3), SciPy library (v 0.18.1) and pandas (v 0.19.2) were utilized for peak finding and curve fitting algorithm. An exponentially modified Gaussian (Foley and Dorsey, 1984) was used as the peak shape function and the desorption temperature values of the peaks as initial guesses for curve fitting. The single thermogram attained initially has a background fitted to the peak to reduce instrumental noise error on the desorption profile integration and then a mathematical fit is applied which is utilised to attribute either a single or multiple desorption profile in which the $T_{max}$ can be retrieved.

**2.3 FIGAERO-CIMS for Vapour Pressure Measurements**

Previous VP measurements have revealed discrepancies in vapour pressures between instruments that differ between compounds depending on the functional groups they contain. In such previous studies it has not proven straightforward to attribute low or high biases to a particular technique, as shown in the Huisman et al., (2013) study. In the following analysis it is assumed there are no functional group or compound specific dependencies applicable to the FIGAERO, for either the PEG, single components or unknown compound analysis. This work also makes the necessary assumption that this filter-based measurement in an uncharacterized mixed matrix yields sub-cooled liquid VPs.

The methodology for retrieving vapour pressures we present in this paper may be subject to some biases when applied to complex chemical systems and this needs to be borne in mind when interpreting results. When measuring thermograms of multi-component systems collected on the FIGAERO, the desorption profiles can exhibit double and/or non-Gaussian peak shapes, often explained by decomposition of higher molecular weight compounds. The thermal decomposition of higher molecular weight compounds can certainly generate errors in the FIGAERO-CIMS $T_{max}$ measurements. This is because any lower molecular weight fragments generated by decomposition will exhibit $T_{max}$ values representative of the $T_{max}$ of the higher molecular weight molecule from which the fragment was generated. Furthermore, inherent to mass spectrometry and CIMS, whilst the molecular composition can be determined, the molecular structure is not known and assumptions have to be made based on likely functional groups present in the system (chamber or environment) that is being measured. Recent studies (Booth *et al*., 2012; Bannan *et al*., 2017; Dang *et al*., 2018) have shown how subtle differences in molecular structure have a significant impact upon vapour pressure. Booth *et al*., (2012), for example, measured the role of ortho, meta, para isomerism in measured solid state and derived sub-cooled liquid vapour pressures of substituted benzoic acids and observed variations of up to 3 order magnitude as a function of this isomerism. Such isomers cannot be differentiated with the CIMS and therefore the assignment of measured $T_{max}$ of compounds with this functional group positioning effect could be dubious and provide broadening or additional peaks, thus affecting the definition of the $T_{max}$ and our methodology presented here.

**2.4 Chamber Experiments**

In addition to the PEG VP calibrations, we also performed FIGAERO measurements of secondary organic aerosols generated in the Manchester Photochemical Aerosol Chamber and vapor pressures of several organic acids are reported here. Briefly, the chamber consists of an 18 m³ Teflon bag illuminated by a bank of halogen lamps and two 6 kW Xenon arc lamps simulating the solar spectrum (further details can be found in Alfarra et al., (2012, 2013). The air charge in the bag was dried





and filtered for gaseous impurities and particles, prior to humidification with high purity de-ionised water. The biogenic SOA precursor α-pinene was injected into the chamber with an initial mixing ratio of 125 ppb. $NO_x$ was added with initial mixing ratios of 30 ppb. The relative humidity was 40% and the temperature was 25°C.

Gas phase measurements were made from the chamber through a 0.75 m long PTFE 6.5 mm OD unheated inlet drawn at 2.2 SLM. Particles were collected through a 1.0 m stainless steel 6.2 mm OD inlet at a flow rate of 2 SLM. The same procedure for obtaining the filter background and the same thermal desorption cycle as used in the UMan FIGAERO-CIMS PEG experiments were utilised for the chamber experiments. In addition a chamber blank was then taken under the same conditions as the sample run on that day, where a 45 minute collection on the filter and subsequent desorption was
completed in a chamber with no VOC added and with no detectable particles in the chamber. The chamber experiments were performed using a 45 minute trace gas analysis and collection on to the PTFE filter.

## 3. Results

### 3.1 The relationship between VP and $T_{max}$

Thermograms are shown in Figure 3 for the PEG samples as measured by the UMan FIGAERO-CIMS, from which the $T_{maxes}$ are retrieved in a process described above. $T_{max}$ values for the PEG compounds are summarized in Table 1 where we also compare our determinations against literature VP measurements reported VPs at 298 K for the PEG series (Krieger *et al*.
2018), and illustrated in figure 4. The vapour pressure range of the PEG series covers and extends, an atmospherically relevant range between 1 and $10^{-4}$ Pa, where compounds with $P_{298K}>1$Pa exist entirely in the gas phase under atmospherically reasonable conditions and compounds with $P_{298K} <10^{-4}$ Pa will exist largely in the particle phase (Valorso et al., 2011). This range of compounds allows characterisation of the FIGAERO across the range of volatilities that are most important throughout the lower atmosphere.
A single exponential fit to the data on the VP at 298 K derived from the PEG series and extracted $T_{max}$ can provide a relationship between $T_{max}$ and VP:

$$VP \ (Pa) = 0.2612exp^{-0.071Tmax}, \ with \ T_{max} \ in \ (^OC). \qquad (1)$$
### 3.2 Evaluating the VP calibration for FIGAERO using single compounds with known VP

The PEG VP calibration can be used to derive the VP of other compounds measured by the FIGAERO ToF-CIMS by extracting the $T_{max}$ of compounds and applying equation 1 to the measured value. By choosing a range of compounds with
known and characterized VP the calibration can be evaluated and may then be utilized for compounds of unknown VP that can be measured with the CIMS.

By way of validation Table 2 and Figure 5 show laboratory single component measurements of $T_{max}$, for a variety of carboxylic acid species, alongside both literature values of their vapour pressure and their calculated vapour pressures using
the PEG calibration curve. Whilst these measurements come from both the UMan and GU FIGAERO-CIMS, the PEG series was not measured by the GU FIGAERO-CIMS. Therefore the same calibration function, derived from the UMan CIMS, is utilised for other instruments. Table 3 and Figure 6 report extracted $T_{max}$ and calculated VPs from a chamber experiment in the Manchester Aerosol Photochemical Chamber using the UMan FIGAERO-CIMS. Where possible the recommended VP values from the Bilde *et al*., (2015) study are used for comparison, as these are the best available literature values available
other than the PEG series.

Using the PEG series calibration for single component measurements it is clear from Table 2 that there is a very good agreement between the FIGAERO and literature vapour pressures. The measurements from the chamber also show a good agreement, with an average overestimation of 67%, which is still well within the reported error of instruments such as the
KEMS in the subcooled liquid state (Booth *et al*., 2012). Figures 5 and 6 show that for the compounds presented by Bilde et al., (2015), as well as selected others, the GU and UMan $T_{max}$ extracted VPs agree very well with the literature. This shows that the PEG series calibration could potentially be applied for different instruments and different reagent ions, depending of course on the configuration and generation of FIGAERO that is being used. Nevertheless, calibration of individual FIGAERO inlets is highly recommended as small changes in the position of the thermocouple, contact time with the heater
and nitrogen, nitrogen flow rate, surface area of the filter among other factors can affect thermograms.

## 4. Discussion and Outlook

We present here the calibration of two FIGAERO inlets coupled to the ToF-CIMS for extracting volatility information from
single component and chamber measurements. Recent comparison of atmospheric component vapour pressures (Krieger et al., 2018) has identified the PEG series as a group of compounds that can be trusted as reference compounds for a range of measurement methods that across the full range of tropospherically representative vapour pressures. This paper shows that this series can be used to calibrate the vapour pressure of single components using the FIGAERO inlet coupled to ToF-CIMS. We have evaluated the derived vapour pressures against a wider range of atmospherically relevant single compounds





and compounds identified in chamber oxidation experiments that have a known vapour pressure and demonstrate consistency with other VP techniques. This offers a pathway to determining VPs from FIGAERO-ToF-CIMS for the many atmospheric compounds that are not yet characterised.

We do note that the FIGAERO is not interference free, mixtures affecting single component VPs and state differences in mixed component systems will affect retrieved VPs especially when organic aerosol concentrations are high. Despite the seemingly good agreement with the UMan and GU FIGAERO for the measurements reported here, it is necessary to independently calibrate each FIGAERO inlet, especially when using a generation of FIGAERO different to the commercially available inlet. The authors believe that such single component measurements of reference compounds, most accurately and

confidently using the PEG series, are essential for understanding the extracted information from the FIGAERO, and other VP measurement techniques, in order to better understand the atmospheric implications of such measurements.

The stability of the PEGs allowed sharing of samples to ensure the same quality between the institutions as those that participated in the Krieger *et al*., (2018) study. Samples should be stored, handled and measured on the same time scale to

reduce as much as possible the chance of contamination. We propose that the same procedure could be undertaken to run an inter-comparison between different FIGAERO inlets.

### 5. Acknowledgements

This work was part funded under NERC grant NE/N013794/1. JTJ acknowledges the U.S. Department of Energy Small Business Innovative Research program (award number DE-SC0004577) that funded the development and commercialization of the CIMS-FIGAERO system reported here and the supporting collaboration with Felipe Lopez-Hilfiker and Joel Thornton from University of Washington.

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

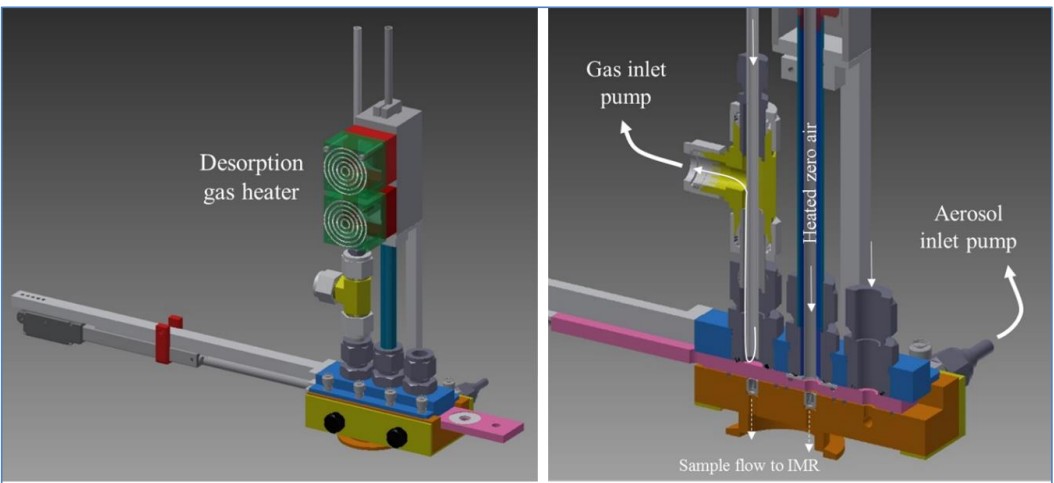


**Figure 1: Drawing of the ARI FIGAERO assembly. Panel on left shows full assembly with mechanical actuator that controls gas sampling/aerosol collection or aerosol desorption operating modes. Panel on right is a cross-sectional view that show flows for both gas and particle sampling mode and the two apertures that connect with the IMR. In this view the FIGAERO slide is positioned in the aerosol desorption mode and the gas sample flow into the IMR is**

**closed.**



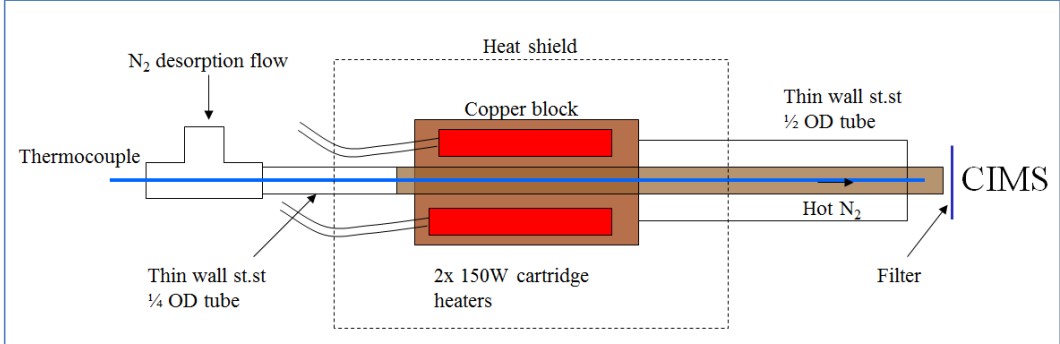

**Figure 2: internal schematic of the FIGAERO desorption gas heater unit. Temperature above the filter is measured at the end point of the heated tube by the long thermocouple running through the inlet, shown here in blue.**

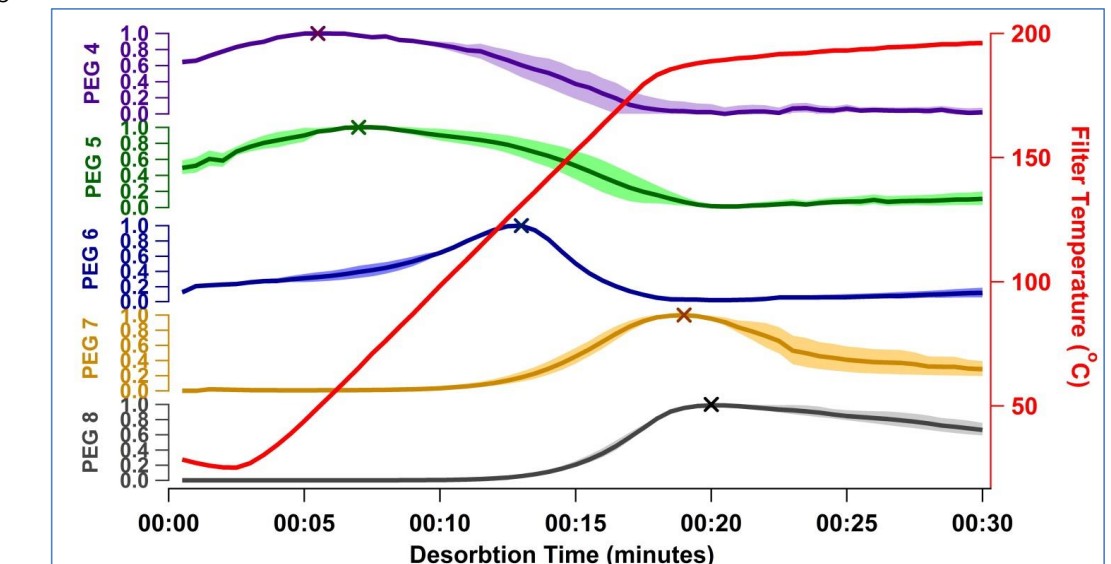

**Figure 3: Thermograms from the PEG series as detected by the FIGAERO-CIMS employing Iodide adduct ionization, all product ion intensities are normalised to 1. Thick coloured lines show the mean of the thermograms and the associated shaded areas show the standard deviation of the 4 thermograms. Crosses show the extracted $T_{max}$.**
10 **Red line shows the temperature just above the filter.**





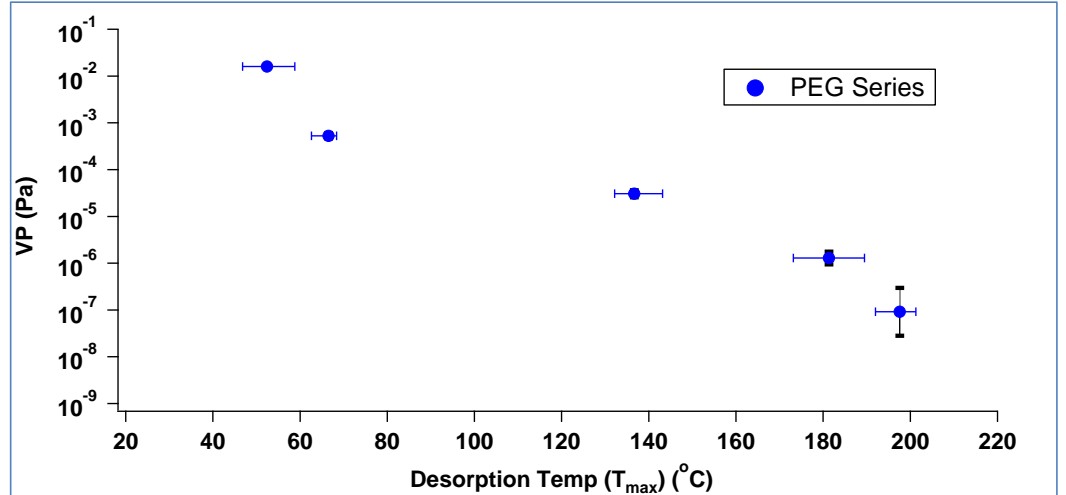

**Figure 4: Reported vapour pressure measurements of the PEG series (4-8) and associated T$_{max}$ values extracted from the UMan FIGAERO-CIMS. Errors on the y-axis are those reported in the Kreiger et al., (2018) study. Errors in the T$_{max}$ (x-axis) are the maximum variation seen within the 4 thermograms from which the mean value was derived.**

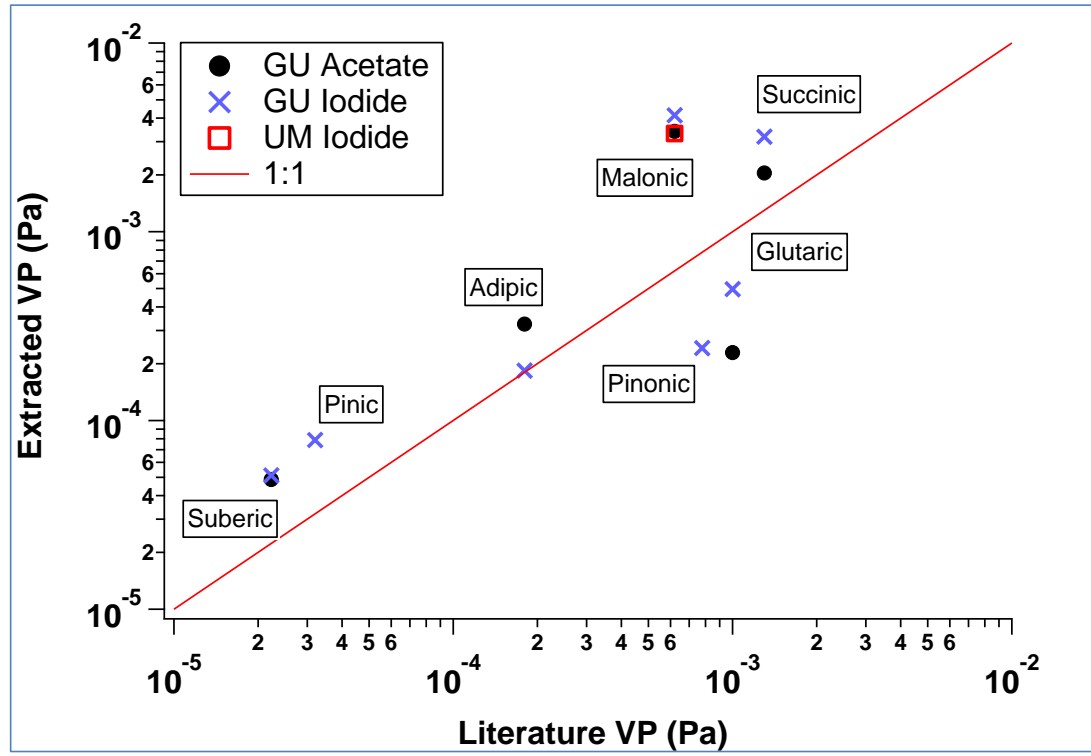

**Figure 5: Extracted VP from the UMan and GU FIGAERO-CIMS, plotted against reported subcooled saturation vapour pressures from the literature, through utilization of the PEG calibration. These measurements are made using single compounds from the UMan and GU FIGAERO-CIMS (see Table 3).**



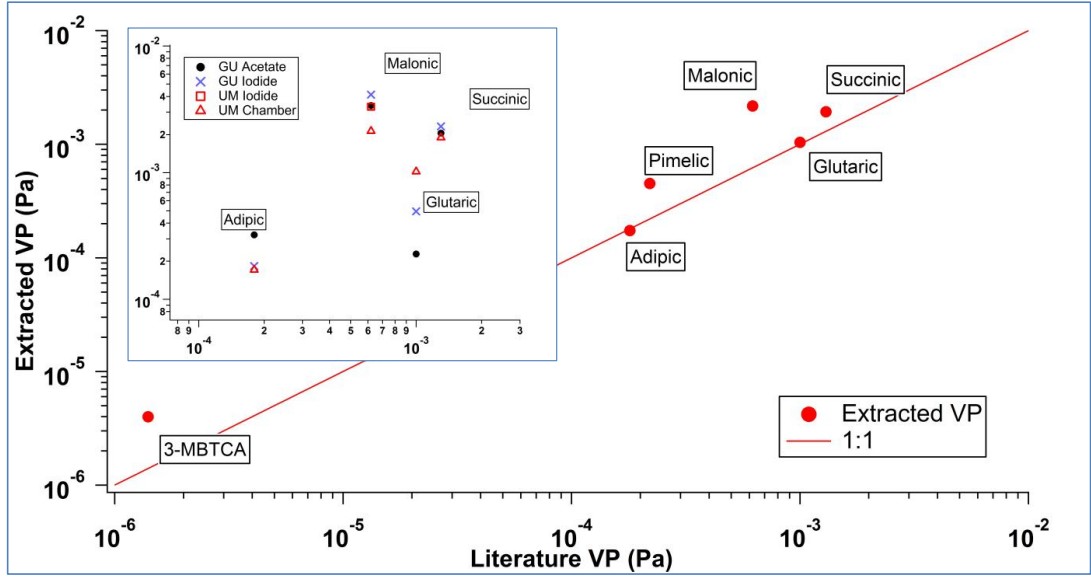

**Figure 6: Extracted VP from the UMan FIGAERO plotted against reported VPs from the literature. These measurements are made using the UMan instrument from the Manchester SOA chamber experiments (see Table 3). Direct comparisons are made for adipic, glutaric, malonic and succinic acids, which were measured in both the single component and chamber study, shown on the inset panel, axis are as described in the main figure.**

**Table 1: Reported vapour pressure measurements of the PEG series (4-8) at 298.15 K (Krieger _et al._, 2018) and associated $T_{max}$ values extracted from the UMan FIGAERO- CIMS. All $T_{max}$ values are an average of 4 individual thermograms for each PEG sample. Errors in the $T_{max}$ are the maximum variation seen within the 4 thermograms.**

| PEG | VP (Pa) | $T_{max}$ ($^O$C) |
|---|---|---|
| **4** | $1.69 \,^{+0.11}_{-0.10}$ x10$^{-2}$ | $52.4 \,^{+6.4}_{-5.6}$ |
| **5** | $5.29 \,^{+0.75}_{-0.65}$ x10$^{-4}$ | $66.5 \,^{+1.8}_{-3.9}$ |
| **6** | $3.05 \,^{+0.59}_{-0.49}$ x10$^{-5}$ | $136.6 \,^{+6.5}_{-4.5}$ |
| **7** | $1.29 \,^{+0.48}_{-0.35}$ x10$^{-6}$ | $181.3 \,^{+8.1}_{-8.1}$ |
| **8** | $9.20 \,^{+20.4}_{-6.4}$ x10$^{-8}$ | $197.5 \,^{+3.7}_{-5.6}$ |

**Table 2: Extracted $T_{max}$ values and calculated VPs through utilization of the PEG calibration compared against literature data of subcooled saturation vapour pressures. A single component measurement is defined as a single calibration compound being placed on the filter and desorbed as per the description in the methods.**

| Compound | Detected As | Tmax (°C) | Reagent Ion and Instrument | FIGAERO VP (Pa) | Literature VP (Pa) | Source |
|---|---|---|---|---|---|---|
| **Malonic** | I.C$_3$H$_4$O$_4$- | 61.5 | UMan Iodide | 3.32X10$^{-3}$ | 6.20X10$^{-4}$ | Bilde et al., 2015 |
| | I.C$_3$H$_4$O$_4$- | 58.4 | GU Iodine | 4.13X10$^{-3}$ | 6.20X10$^{-4}$ | Bilde et al., 2015 |
| | C$_3$H$_3$O$_4$- | 61.2 | GU Acetate | 3.39X10$^{-3}$ | 6.20X10$^{-4}$ | Bilde et al., 2015 |
| **Succinic** | I.C$_4$H$_6$O$_4$- | 62.1 | GU Iodine | 3.18X10$^{-3}$ | 1.30X10$^{-3}$ | Bilde et al., 2015 |
| | C$_4$H$_5$O$_4$- | 68.3 | GU Acetate | 2.05X10$^{-3}$ | 1.30X10$^{-3}$ | Bilde et al., 2015 |
| **Glutaric** | I.C$_5$H$_8$O$_4$- | 88.3 | GU Iodine | 4.95X10$^{-4}$ | 1.00X10$^{-3}$ | Bilde et al., 2015 |
| | C$_5$H$_7$O$_4$- | 99.2 | GU Acetate | 2.28X10$^{-4}$ | 1.00X10$^{-3}$ | Bilde et al., 2015 |
| **Adipic** | I.C$_6$H$_{10}$O$_4$- | 102.3 | GU Iodine | 1.83X10$^{-4}$ | 1.80X10$^{-4}$ | Bilde et al., 2015 |
| | C$_6$H$_9$O$_4$- | 94.3 | GU Acetate | 3.23X10$^{-4}$ | 1.80X10$^{-4}$ | Bilde et al., 2015 |
| **Suberic** | I.C$_8$H$_{14}$O$_4$- | 120.3 | GU Iodine | 5.10X10$^{-5}$ | 2.23X10$^{-5}$ | Booth et al., 2011 |
| | C$_8$H$_{13}$O$_4$- | 121 | GU Acetate | 4.85X10$^{-5}$ | 2.23X10$^{-5}$ | Booth et al., 2011 |
| **Pinonic** | I.C$_{10}$H$_{16}$O$_3$- | 98.4 | GU Iodine | 2.41X10$^{-4}$ | 7.78X10$^{-4}$ | Booth et al., 2011 |



| Pinic | I.C$_9$H$_{14}$O$_4$- | 114.2 | GU Iodine | 7.86X10$^{-5}$ | 3.20X10$^{-5}$ | Bilde and Pandis 2001 |
|---|---|---|---|---|---|---|

**Table 3: Extracted T$_{max}$ values and VP at 298k from chamber SOA experiments as measured by the UMan Iodide FIGAERO-CIMS. T$_{max}$ are an average measured over the 7 desorptions (not including background) from the chamber experiment.**

| Compound | Detected As | Tmax (°C) | FIGAERO VP (Pa) | Comparison VP (Pa) | Source |
|---|---|---|---|---|---|
| **Malonic** | I.C$_3$H$_4$O$_4$- | 67.5 | 2.17X10$^{-3}$ | 6.20X10$^{-4}$ | Bilde et al., 2015 |
| **Succinic** | I.C$_4$H$_6$O$_4$- | 69.1 | 1.93X10$^{-3}$ | 1.30X10$^{-3}$ | Bilde et al., 2015 |
| **Glutaric** | I.C$_5$H$_8$O$_4$- | 77.8 | 1.04X10$^{-3}$ | 1.00X10$^{-3}$ | Bilde et al., 2015 |
| **Adipic** | I.C$_6$H$_{10}$O$_4$- | 103.0 | 1.74X10$^{-4}$ | 1.80X10$^{-4}$ | Bilde et al., 2015 |
| **Pimelic Acid** | I.C$_7$H$_{12}$O$_4$- | 89.6 | 4.51X10$^{-4}$ | 2.20X10$^{-4}$ | Bilde et al., 2015 |
| **3-MBTCA** | I.C$_8$H$_{12}$O$_6$- | 156.2 | 3.99X10$^{-6}$ | 1.50X10$^{-6}$ | Lienhard et al., 2015 |