# Peer review of "A method for extracting calibrated volatility information from the FIGAERO-HR-ToF-CIMS and its experimental application"

_Atmospheric Measurement Techniques, 2018_

## Referee Comment (RC1) · Anonymous Referee #1 · 7 Sep 2018

General comments:

The manuscript "A method for extracting calibrated volatility information from the FIGAERO-HR-ToF-CIMS and its application to chamber and field studies" by Thomas J. Bannan et al. reports on well-executed experiments that succeeded well in calibrating the desorption heating of FIGAERO-HR-TOF-CIMS instruments. FIGAERO is a fairly novel technique that has started to gain wide-spread use recently and proven powerful in retrieving information on composition, volatility, and more, from organic aerosol particles. Therefore, I think that publication of this manuscript in Atmospheric

[Figure]

Measurement Techniques (AMT) will be very useful for the atmospheric science community. The authors describe their measurements well, and convinced this reviewer that they have been carried out properly and with due diligence. I recommend publication of this manuscript in AMT, however, following some minor revisions.

First, the title includes "application to chamber and field studies". However, I believe that application was only done to chamber experiments, so I suggest to remove the reference to field studies. (Although the method can certainly be applied to field studies, but if that is the intention to communicate, it may be better to reformulate.)

More importantly, a few parts of the manuscript are not as clearly described or as clearly discussed as I would prefer. And I think that some desirable information is missing regarding experimental details. I will discuss these issues in detail in the following.

Detailed comments:

Regarding the discussion of blank or background measurements (both for calibration and chamber SOA experiments, sections 2.2 and 2.4): Was there noticeable blank (=background?) signal, and was there a need for subtracting from data, or how where blank measurements treated?

p. 1, l. 55-56: Many LVOC and all ELVOC, probably very relevant for SOA, are typically considered to have much lower vapor pressures (e.g. Tröstl et al., 2016). (That is, referring to room-temperature saturation vapor pressures.) So I suggest revising that statement.

p. 3, l. 43: Does "inside" mean inside the heating block?

p. 3, l. 63: Do I understand correctly that the filter was replaced after the blank measurement? Why? More detailed description a few paragraphs below suggests that the filter was already new for the blank measurements and not changed thereafter, but not sure... Suggest reformulating. Better yet, combine, so that the filter handling procedure

is only described once.

p. 4 , l. 10-15: I don't understand the notation "200/300 °C". Later-on a heating to 300 °C is not mentioned again. (And wouldn't PTFE start getting problems when heating that high? Or is the 300 °C referring to the temperature measured farther away from the filter?) Anyway, the heating rate (6.1 °C/min) and time (45 min) is consistent with 300 °C. I assume the equivalent heating rate for standard desorptions was (300-25)/10 = 27.5 °C? Maybe best if the authors could include that default heating rate, and clarify the issue regarding 200 vs. 300 °C.

p. 4, l. 27-32: The peak fitting procedure used for GU FIGAERO data appears quite complex. This paper is maybe a good place to present that procedure more clearly, e.g. by adding an explanatory figure that shows example data and the fits at various stages of the fitting procedure.

Section 3.1: According to section 2, 0.1 $\mu$L of solution were deposited during all calibration experiments. But it would be useful for the community to know also their concentration and the total mass of PEG that was deposited. That information could be included in Table 1, for instance. And please plot that fit (Eq. 1) also in Fig. 4. Regarding the data in Fig. 4: I feel there is a somewhat large variability in the observed Tmax values (if I think about my own experience with FIGAERO data). I would expect better reproducibility, in particular given that the PEG deposits are chemically simple and presumably identical in terms of amounts deposited for individual experiments and deposition technique. Do the authors have ideas what could have caused that variability?

Section 3.2: As commented above regarding section 3.1, please add information about the deposits in those experiment, e.g. in Table 2. And are the Tmax from single measurements or an average over several repetitions? Analogously, Table 3 should include the aerosol mass collected. There has been some indications that the amount collected can affect the observed Tmax (e.g. Huang et al., 2018). It could also be interesting

to know about the variability in observed Tmax, as the listed values are the mean of 7 measurements. In general, the agreement between retrieved vapor pressures and literature values is convincing. But regarding the SOA experiments, I would have expected observed ("effective") vapor pressures to be much lower compared to pure-compound values, due to the Raoult effect (e.g. Donahue et al., 2006): I guess that the various acids reported here respectively only constitute a small fraction of the SOA (by the way, another piece of information that could be reported, in Table 3). Taking into account a Raoult effect would presumably worsen the agreements with literature considerably. I interpret that such that the evaporation from SOA is maybe not directly observed. Instead it might be vapor-pressure controlled processes that follow the initial SOA evaporation that somehow determine Tmax. Interactions with instrument surfaces? Could that explain the large differences reported from different FIGAERO versions (Stark et al., 2017)? I am curious about the authors' opinion on that. (And by the way, that Stark et al. paper should be cited in this manuscript. There isn't too many reports of FIGAERO calibrations out there yet, and that is one of them.)

p. 5, l. 21: I think the authors mean 10ˆ-7 Pa instead of 10ˆ-4 Pa? (I agree with the use of 10ˆ-4 Pa in the next line though.)

p. 5, l. 62: There is some mistake in that first sentence. Besides, as mentioned above, I disagree that the PEG calibration compounds actually cover the full volatility range of atmospherically relevant organics. At least the specific PEG compounds used here.

Minor/technical comments:

p. 2, l. 8-10: double-mention of ongoing measurements, suggest mentioning only once for style

p. 2, l. 28: changes. Same in p. 3, l. 37-38, or maybe I am misreading these sentences.

p. 2, l. 30: "and" too much?

p. 2, l. 35: "," too much

p. 3, l. 26: I assume the GU CIMS was operated with either acetate or iodide reagent ions?

p. 3, l. 38: Definition of Tmax is inconsistent with its definition before (and again at p. 3, l. 48). Probably wrong use of "defined".

p. 4, l., 41: I find the sentence hard to follow

p. 4, l. 64: odd amount of brackets

p. 5, l. 20: unclear meaning of "extend"

References:

Donahue, N. M., Robinson, A. L., Stanier, C. O., and Pandis, S. N.: Coupled Partitioning, Dilution, and Chemical Aging of Semivolatile Organics, Environ. Sci. Technol., 40, 2635-2643, doi:10.1021/es052297c, 2006.

Huang, W., Saathoff, H., Pajunoja, A., Shen, X., Naumann, K. H., Wagner, R., Virtanen, A., Leisner, T., and Mohr, C.: $\alpha$-Pinene secondary organic aerosol at low temperature: chemical composition and implications for particle viscosity, Atmos. Chem. Phys., 18, 2883-2898, doi:10.5194/acp-18-2883-2018, 2018.

Stark, H., Yatavelli, R. L. N., Thompson, S. L., Kang, H., Krechmer, J. E., Kimmel, J. R., Palm, B. B., Hu, W., Hayes, P. L., Day, D. A., Campuzano-Jost, P., Canagaratna, M. R., Jayne, J. T., Worsnop, D. R., and Jimenez, J. L.: Impact of Thermal Decomposition on Thermal Desorption Instruments: Advantage of Thermogram Analysis for Quantifying Volatility Distributions of Organic Species, Environ. Sci. Technol., 51, 8491-8500, doi:10.1021/acs.est.7b00160, 2017.

Tröstl, J., Chuang, W. K., Gordon, H., Heinritzi, M., Yan, C., Molteni, U., Ahlm, L., Frege, C., Bianchi, F., Wagner, R., Simon, M., Lehtipalo, K., Williamson, C., Craven, J. S., Duplissy, J., Adamov, A., Almeida, J., Bernhammer, A.-K., Breitenlechner, M., Brilke,

S., Dias, A., Ehrhart, S., Flagan, R. C., Franchin, A., Fuchs, C., Guida, R., Gysel, M., Hansel, A., Hoyle, C. R., Jokinen, T., Junninen, H., Kangasluoma, J., Keskinen, H., Kim, J., Krapf, M., Kürten, A., Laaksonen, A., Lawler, M., Leiminger, M., Mathot, S., Möhler, O., Nieminen, T., Onnela, A., Petäjä, T., Piel, F. M., Miettinen, P., Rissanen, M. P., Rondo, L., Sarnela, N., Schobesberger, S., Sengupta, K., Sipilä, M., Smith, J. N., Steiner, G., Tomè, A., Virtanen, A., Wagner, A. C., Weingartner, E., Wimmer, D., Winkler, P. M., Ye, P., Carslaw, K. S., Curtius, J., Dommen, J., Kirkby, J., Kulmala, M., Riipinen, I., Worsnop, D. R., Donahue, N. M., and Baltensperger, U.: The role of low-volatility organic compounds in initial particle growth in the atmosphere, Nature, 533, 527-531, doi:10.1038/nature18271, 2016.

———————————————

---

## Referee Comment (RC2) · Anonymous Referee #2 · 12 Sep 2018

General: To understand partitioning of (organic) substances in the atmosphere is a key issue. Recently FIGAERO CIMS was developed as a promising method. However to avoid misinterpretations of field and chamber measurements carefully characterization is needed. This manuscript clearly contributes to such a characterization. It addresses the derivation of VP from the conc. maximum of desorption curve in FIGAERO thermograms, utilizing a well characterized reference set of PEGs. The paper is interesting and compact and well written. I suggest publication in AMT after the authors addressed some minor points below.

[Figure]

Minor points:

p2l7: I am wondering about the Ovadnevaite et al. 2017 reference in the context of gas-phase - particle phase partitioning.

p2l30: What is the reason / are the conditions for observation of Tmax? And related: what is the physics behind the expected (?, Gaussian, p4l47-50) shape of the thermogram? Schobesberger thinks of evaporating particles, but you seem to assume liquid states. Wouldn't the liquids spread and wet the filter fibers? I understand that those details are not really important for the results, but it may help to give an idea/introduction about your imagination of states and processes.

p3l37: Something is wrong with this sentence. Please, check and reformulate.

p4l13f: "200/300" I don't understand what is meant. 45 min x 6.1° will get you 275° on top of the RT of 25°,i.e. to 300°C. You may split and separate the info you intended to give into two sentences.

p4l29: I don't understand what you want to say here. How can subtraction of a background fit improve instrumental noise. I guess you have to extend here a little bit more.

p4l47: Tmax for a fragment only reflects Tmax of the parent compound, if the fragmentation happened after evaporation, i.e. in the gas-phase. However, as far as I understood, there could be also fragmentation - at weak bonds - in the particulate phase, isn't it? Then Tmax of appearance of the fragment does not represent the thermal properties of the parent anymore. If so, you have to modify this statement in the manuscript accordingly.

p5l26: I suggest to include/show this fit in Figure 4.

Figure 3: Sample 1-3 where measured during the ramp. Sample 4,5 after the ramp stopped and the system stabilized (at a lower rate of t increase). Does the ramp rate has any effect on the Tmax? I guess so. But then sample 4,5, where not measured at same condition as sample 1-3. Please comment.

Figure 6, p4l62: I don't understand what is shown here. In the chamber measurement you don't know if the detected formulas are the dicarboxylic acids as tagged? Or did you add the dicarboxylic acids as such. Please, clarify.

Typos etc.

Check co-author name "Krieger" vs "Kreiger"

Check the use of capitals in figure, it should Figure

Check the use of capitals in peg / PEG

p2l10: 'Such measurements "of are" ongoing...'

p3l9: maybe: "(> 6 month)"

p4l26: too many "averages" here

p4l49: "mass spectrometry and CIMS" is somehow double, you may want to modify this phrase

p5l18: "Tmaxes"

p7l12: the "Foley" reference is incomplete
* * *

---

## Referee Comment (RC3) · Anonymous Referee #3 · 22 Sep 2018

The manuscript, 'A method for extracting calibrated ... ' describes a method of using a series of PEG compounds to calibrate the FIGAERO for determining vapor pressures of detected compounds. The work presents a useful concept that can be used by a growing number of research groups that use the FIGAERO and similar techniques to normalize (or standardize) measurements of the volatility of OA components. The manuscript is succinct, which is nice, but some potentially major details are missing. Please address/clarify the below issues that may affect the applicability of the presented concept, after which the manuscript can be considered for publication.

[Figure]

A series of PEG compounds was used as calibrants to connect literature VP values of the PEG compounds to FIGAERO Tmax values. The calibration curve as defined by equation 1 is not shown on figure 4. It would appear to me that a simple exponential curve does not fit the observed values well. Please include the calibration curve on figure 4, and discuss potential reasons for the deviation from the calibration curve.

The PEG compounds on figure 3 exhibit different desorption profiles, that is, some are much broader than others. Why? How were the PEG compounds introduced? Injected? Individually or together? Same heating ramp rate for each PEG? Are the four thermograms of each PEG of the same amount introduced? Have the authors tried injecting widely ranging amounts of a PEG compound? How much does the amount introduced affect the Tmax value? The amount of OA present can affect the Tmax values, as reported by many FIGAERO users. This issue can severely affect the applicability of the presented technique, thus the Tmax dependence on OA loading needs to be addressed carefully. Also, please consider plotting signal versus temperature, not time on figure 3.

I am particularly concerned with the concept of Tmax for compounds like PEG 4, 5, and 8, all of which show very broad desorption profiles. For these species in particular, the amount introduced, heating ramp rate, etc. can affect Tmax values greatly. Perhaps consider reporting the temperature at which half the mass comes off the filter, as opposed to Tmax.

There is brief mention of alpha-pinene oxidation, but no figures are shown and no results discussed. Please elaborate.

minor manually extracted? line 26 page 4

line 46 page 5, "very good"? quantify how good, R value, slope, etc?

line 26 page 2, not Teflon (specific to DuPont product), report specific compound like PFA or PTFE

line 25 page 4, change "form" to from

---

## Author Comment (AC1) · 23 Nov 2018

**A method for extracting calibrated volatility information from the FIGAERO-HR-ToF-CIMS and its application to chamber and field studies**

We thank the reviewers for their time evaluating this manuscript and their positive comments relating to this work. The corrections and additions made as a result of these comments have greatly improved the consistency and focus of this work. The response to each point immediately follows each comment and is coloured red.

**Anonymous Referee #1**

The manuscript "A method for extracting calibrated volatility information from the FIGAERO-HR-ToF-CIMS and its application to chamber and field studies" by Thomas J. Bannan et al. reports on well-executed experiments that succeeded well in calibrating the desorption heating of FIGAERO-HR-TOF-CIMS instruments. FIGAERO is a fairly novel technique that has started to gain wide-spread use recently and proven powerful in retrieving information on composition, volatility, and more, from organic aerosol particles. Therefore, I think that publication of this manuscript in Atmospheric Measurement Techniques (AMT) will be very useful for the atmospheric science community. The authors describe their measurements well, and convinced this reviewer that they have been carried out properly and with due diligence. I recommend publication of this manuscript in AMT, however, following some minor revisions.

**Detailed comments**

First, the title includes "application to chamber and field studies". However, I believe that application was only done to chamber experiments, so I suggest to remove the reference to field studies. (Although the method can certainly be applied to field studies, but if that is the intention to communicate, it may be better to reformulate.)

Response: The paper title has been changed to: "A method for extracting calibrated volatility information from the FIGAERO-HR-ToF-CIMS and its experimental application"

Regarding the discussion of blank or background measurements (both for calibration and chamber SOA experiments, sections 2.2 and 2.4): Was there noticeable blank (=background?) signal, and was there a need for subtracting from data, or how where blank measurements treated?

Response: for the calibration there was no background signal in each thermogram which needed to be analysed. A more detailed background procedure for the chamber experiments is now also included. However, given that quantitative concentration data is not reported, only the behaviour of the desorption profiles, there was no need for subtracting backgrounds from the data reported here.

*"First, a new filter was placed in the FIGAERO and the temperature was ramped to 200°C for 10 minutes to ensure the filter was clean and then cooled. A ramp, soak and cool cycle matching that of the subsequent sample was then completed to obtain the filter background. In addition, and after the filter background, a chamber background was then taken daily that involved a 45 minute filter collection of air from the chamber with no VOC added and with no detectable particles in the chamber and subsequent desorption."*

p. 3, l. 63: Do I understand correctly that the filter was replaced after the blank measurement? Why? More detailed description a few paragraphs below suggests that the filter was already new for the blank measurements and not changed thereafter, but not sure... Suggest reformulating. Better yet, combine, so that the filter handling procedure is only described once.

Response: The filter used for the blank desorption was used for the following PEG experiment and this has been now better explained in the text. Sections 2.3 and 2.4 have now been switched for better flow of the experimental procedures.

*"Prior to each sample measurement being made using both the GU and UMan FIGAERO-CIMS instruments, background measurements were obtained. First, a new filter was placed in the FIGAERO and the temperature was ramped to 200°C for 10 minutes to ensure the filter was clean and then cooled. A ramp, soak and cool cycle matching that of the subsequent sample was then completed to obtain the background. During the PEG series measurements the filter was ramped to 200°C (temperature above the filter) over a period of 20 minutes (at a rate of 8.75°C min$^{-1}$), held at 200°C for 10 minutes and finally allowed to cool back to room temperature for a period of 5 minutes. The same cycle was used for the single component measurements for both the GU and UMan instruments. It is however noted that the analysis provided here does not take into account the possibility of a change in ramp rate affecting the $T_{max}$. It is therefore recommended that the calibration cycles match that of the measurements. Temperature cycles and gas flows were controlled using the ARI EyeOn™ control system."*

p. 1, l. 55-56: Many LVOC and all ELVOC, probably very relevant for SOA, are typically considered to have much lower vapor pressures (e.g. Tröstl et al., 2016). (That is, referring to room-temperature saturation vapor pressures.) So I suggest revising that statement.

Response: That is correct – the sentence has been rewritten as;

*"Functionalization can create compounds with a huge range of expected saturation vapour pressures between 0.1 Pa and 10$^{-10}$ Pa and lower (Jimenez et al., 2009; O'Meara et al., 2014; Bilde et al., 2015, Tröstl et al., 2016)."*

p. 3, l. 43: Does "inside" mean inside the heating block?

Response: as detailed in the text it relates to inside the copper tube, however to be clearer a link to Figure 2 is added where the position of the thermocouple in question is illustrated

*"The gas temperature is measured by a 1/16" diameter thermocouple positioned inside and just near the exit of the ¼" OD copper tube (~5 mm above the PTFE Teflon filter as detailed in Figure 2)."*

p. 4 , l. 10-15: I don't understand the notation "200/300 ∘C". Later-on a heating to 300 ∘C is not mentioned again. (And wouldn't PTFE start getting problems when heating that high? Or is the 300 ∘C referring to the temperature measured farther away from the filter?) Anyway, the heating rate (6.1 ∘C/min) and time (45 min) is consistent with 300 ∘C. I assume the equivalent heating rate for standard desorptions was (300-25)/10 = 27.5 ∘C?

Response: the notation was here to show that some desorptions during the experiments were run to 200$^{0}$C and others to 300$^{0}$C, but we agree that this is not clear. The hotter filter temperature and longer desorption times were used, as described in the text, to enable resolution of multiple peak desorptions and calculation of more accurate $T_{max}$ values, but as all the results presented in this study are based on the measurements up to 200$^{0}$C, references to the longer desorption and hotter temperatures have been removed as we agree that at 300$^{0}$C the filters would likely not be thermally stable.

The position of the temperature that is measured is now also clarified in the text

*"During the PEG series measurements the filter was ramped to 200°C (temperature above the filter) over a period of 20 minutes (at a rate of 8.75°C min$^{-1}$), held at 200°C for 10 minutes and finally allowed to cool back to room temperature for a period of 5 minutes."*

Maybe best if the authors could include that default heating rate, and clarify the issue regarding 200 vs. 300 ∘C.

Response: agreed. This has now been included.

p. 4, l. 27-32: The peak fitting procedure used for GU FIGAERO data appears quite complex. This paper is maybe a good place to present that procedure more clearly, e.g. by adding an explanatory figure that shows example data and the fits at various stages of the fitting procedure.

Response: A substantially expanded and improved description of the details of this method has now been included in the supplementary information for this paper.

Section 3.1: According to section 2, 0.1 µL of solution were deposited during all calibration experiments. But it would be useful for the community to know also their concentration and the total mass of PEG that was deposited. That information could be included in Table 1, for instance.

Response: the total mass and concentrations of PEG used in this study are now discussed in the following text:

*"Four desorptions of each PEG were performed by depositing 0.1 µl of two different concentrations (two repeats of each), of 200 µg cm$^{-3}$ and 2000 µg cm$^{-3}$, with a mean of the 4 desorptions being reported as the $T_{max}$. No linear dependence of $T_{max}$ with concentration was observed across this concentration range. As with any calibration it is recommended to use a comparable amount of calibrant material as would be expected to accumulate during the measurements, as it is noted that the amount of condensed material on the filter can affect the $T_{max}$. A range of calibration concentrations larger than that reported in this study is suggested for future studies and the small range is noted here as a limitation of this study. PEG calibrations were generally conducted individually and were manually syringed on to the filter. The reported $T_{max}$ value for the one of highest concentration runs for PEG 4 and PEG 6 as well as PEG 5 and PEG 8 were mixed in two separate experiments. The conditions were designed to ensure that the $T_{max}$ of the PEG series was not mixture dependent, although a more detailed study is required."*

And please plot that fit (Eq. 1) also in Fig. 4.

Response: this has now been added.

Regarding the data in Fig. 4: I feel there is a somewhat large variability in the observed Tmax values (if I think about my own experience with FIGAERO data). I would expect better reproducibility, in particular given that the PEG deposits are chemically simple and presumably identical in terms of amounts deposited for individual experiments and deposition technique. Do the authors have ideas what could have caused that variability?

Response: We have thoroughly checked all known parameters that may affect the thermograms reported here and whilst we recognise there is rather more variability than may be typical for FIGAERO calibrations we are unsure of the reason. In general, varying concentrations and ramp rates might potentially lead to varying desorption profiles; although in this study we did not perturb ramp rates and, with the concentrations used here we see no direct evidence of this. Given the variability between the past reported responses to VP and Tmax, as reported in Stark et al., (2017) the importance of calibrating individual instruments is reiterated, as many effects noted above can affect the thermograms.

Section 3.2: As commented above regarding section 3.1, please add information about the deposits in those experiment, e.g. in Table 2.

Response: As above, this has now been addressed.

And are the Tmax from single measurements or an average over several repetitions?

Response: The Tmax is a mean over 4 repetitions. This is now stated in the text in section 3.2 where the concentrations of each desorption are now also noted.

Analogously, Table 3 should include the aerosol mass collected. There has been some indications that the amount collected can affect the observed Tmax (e.g. Huang et al., 2018).

The variability for the Tmax during this experiment is now reported in table 3 as requested below. We see no relationship between the Tmax and the total mass measured in the chamber, but we agree that this may be an important factor to consider, and is now discussed as below.

*"As with any calibration it is recommended to use a comparable amount of calibrant material as would be expected to accumulate during the measurements, as it is noted that the amount of condensed material on the filter can affect the $T_{max}$. A range of calibration concentrations larger than that reported in this study is suggested for future studies and the small range is noted here as a limitation of this study."*

The range of total mass observed in the chamber during this measurement period to show that this has been considered in the caption of Figure 6.

It could also be interesting to know about the variability in observed Tmax, as the listed values are the mean of 7 measurements.

Response: the maximum observed variability in the Tmax from each of the 7 thermograms is now included in table 3.

In general, the agreement between retrieved vapor pressures and literature values is convincing. But regarding the SOA experiments, I would have expected observed ("effective") vapor pressures to be much lower compared to pure-compound values, due to the Raoult effect (e.g. Donahue et al., 2006): I guess that the various acids reported here respectively only constitute a small fraction of the SOA (by the way, another piece of information that could be reported, in Table 3). Taking into account a Raoult effect would presumably worsen the agreements with literature considerably. I interpret that such that the evaporation from SOA is maybe not directly observed. Instead it might be vapor-pressure controlled processes that follow the initial SOA evaporation that somehow determine Tmax. Interactions with instrument surfaces? Could that explain the large differences reported from different FIGAERO versions (Stark et al., 2017)? I am curious about the authors' opinion on that.

As we have already stated in the paper;

*"This work also makes the necessary assumption that this filter-based measurement in an uncharacterized mixed matrix yields single component sub-cooled liquid VPs."*

And therefore the FIGAERO does not produce a mole fraction scaled vapour pressure. We assume that all components, even if they were originally associated with an aqueous solution when in the aerosol, will have precipitated out of solution according to their solubility as the water is driven off and then evaporate as "pure" components. Each compound therefore exhibits their pure component vapour pressure and it will not depend on their mole (or mass) fractions.

The ARI FIGAERO inlet used in this work is similar to that described by Lopez-Hilfiker et al. (2014), but not identical and differences in the position of the thermocouple position, flows and general configuration of the FIGAEROs will have implications for the reported Tmax values from each instrument. This discussion is however outside the scope of the paper as we are presenting a method of calibration for each user and not the direct translation of results.

And by the way, that Stark et al. paper should be cited in this manuscript. There isn't too many reports of FIGAERO calibrations out there yet, and that is one of them.

Response: Agreed, this paper is now referenced serval times throughout the revised manuscript.

p. 5, l. 21: I think the authors mean 10ˆ-7 Pa instead of 10ˆ-4 Pa? (I agree with the use of 10ˆ-4 Pa in the next line though.)

Response: That is correct; this has been corrected in the text.

p. 5, l. 62: There is some mistake in that first sentence. Besides, as mentioned above, I disagree that the PEG calibration compounds actually cover the full volatility range of atmospherically relevant organics. At least the specific PEG compounds used here.

Response: There was a mistake in that sentence, yes. This has been corrected with an amendment to show that this calibration does not cover the full volatility range of atmospherically relevant organics, but a very significant part of it.  Amended to:

*"Recent comparison of vapour pressure measurement techniques (Krieger et al., 2018) has identified the PEG series as a group of  compounds that can be trusted as reference compounds for a range of measurement methods that  across a wide range of tropospherically representative vapour pressures."*

**Minor/technical comments**

p. 2, l. 8-10: double-mention of ongoing measurements, suggest mentioning only once for style. Response: Completed

p. 2, l. 28: changes. Same in p. 3, l. 37-38, or maybe I am misreading these sentences. Response: change is the correct use here.

p. 2, l. 30: "and" too much? Response: this has been left as is.

p. 2, l. 35: "," too much Response: this has been left as is.

p. 3, l. 26: I assume the GU CIMS was operated with either acetate or iodide reagent ions? Response: this is correct and has been clarified in the text.

*"GU CIMS was operated with acetate or iodide as the reagent ion."*

p. 3, l. 38: Definition of Tmax is inconsistent with its definition before (and again at p. 3, l. 48). Probably wrong use of "defined". Response: yes, defined is the wrong use here. This has been change to *"measured"* in reference to the position that the $T_{max}$ is measured.

p. 4, l., 41: I find the sentence hard to follow Response: this sentence has now been changed slightly.

p. 4, l. 64: odd amount of brackets Response: this has been corrected.

p. 5, l. 20: unclear meaning of "extend" Response: extend has been removed here

**Anonymous Referee #2**

**General**

To understand partitioning of (organic) substances in the atmosphere is a key issue. Recently FIGAERO CIMS was developed as a promising method. However to avoid misinterpretations of field and chamber measurements carefully characterization is needed. This manuscript clearly contributes to such a characterization. It addresses the derivation of VP from the conc. maximum of desorption curve in FIGAERO thermograms, utilizing a well characterized reference set of PEGs. The paper is interesting and compact and well written. I suggest publication in AMT after the authors addressed some minor points below.

**Minor points**

p2l7: I am wondering about the Ovadnevaite et al. 2017 reference in the context of gas-phase - particle phase partitioning.

Response: this has been removed from the text

p2l30: What is the reason / are the conditions for observation of Tmax? And related: what is the physics behind the expected (?, Gaussian, p4l47-50) shape of the thermogram? Schobesberger thinks of evaporating particles, but you seem to assume liquid states. Wouldn't the liquids spread and wet the filter fibers? I understand that those details are not really important for the results, but it may help to give an idea/introduction about your imagination of states and processes.

Response: the state of the material is discussed in reference to the comments of reviewer 1.

p3l37: Something is wrong with this sentence. Please, check and reformulate.

Response: this has been completed

*"The evolution of the MS signals from different compounds the filter is exposed to during the Temperature Ramp phase change independently as a function of temperature creating thermograms that are is m/z specific."*

p4l13f: "200/300" I don't understand what is meant. 45 min x 6.1∘ will get you 275∘ on top of the RT of 25∘ ,i.e. to 300∘C. You may split and separate the info you intended to give into two sentences.

Response: this has now been clarified, as per the request of reviewer 1. Reference to the $300^0$C measurements have now been removed as no data from these measurements from these experiments have been used in the study.

p4l29: I don't understand what you want to say here. How can subtraction of a background fit improve instrumental noise. I guess you have to extend here a little bit more.

Response: correct. In response to this and the comments of reviewer 1 a significant improvement in the details of this method has now been included in the supplementary information.

p4l47: Tmax for a fragment only reflects Tmax of the parent compound, if the fragmentation happened after evaporation, i.e. in the gas-phase. However, as far as I understood, there could be also fragmentation - at weak bonds - in the particulate phase, isn't it? Then Tmax of appearance of the fragment does not represent the thermal properties of the parent anymore. If so, you have to modify this statement in the manuscript accordingly.

Response: this is correct and should also be included. The following has been added to account for this;

*"There also may be fragmentation of weak bonds in the particulate phase, also giving an unrepresentative $T_{max}$ and desorption profile."*

p5l26: I suggest to include/show this fit in Figure 4.

Response: this has now been added.

Figure 3: Sample 1-3 where measured during the ramp. Sample 4,5 after the ramp stopped and the system stabilized (at a lower rate of t increase). Does the ramp rate has any effect on the Tmax? I guess so. But then sample 4,5, where not measured at same condition as sample 1-3. Please comment.

This variability is now noted in the text and is noted as a limitation in the study

*"The same cycle was used for the single component measurements for both the GU and UMan instruments. It is however noted that the analysis provided here does not take into account the possibility of a change in ramp rate affecting the $T_{max}$. It is therefore recommended that the calibration cycles match that of the measurements."*

Figure 6, p4l62: I don't understand what is shown here. In the chamber measurement you don't know if the detected formulas are the dicarboxylic acids as tagged? Or did you add the dicarboxylic acids as such. Please, clarify.

Response: measurements reported here are the dicarboxylic acids measured during the chamber experiments, this has been made clearer in the text now.

*"In addition to the PEG VP calibrations, we also performed FIGAERO measurements of secondary organic aerosols generated in the Manchester Photochemical Aerosol Chamber and vapor pressures of several organic acids (mass accuracy all <2 ppm) from measurements made in these experiments are reported here."*

Check co-author name "Krieger" vs "Kreiger"

Response: the manuscript has been thoroughly checked for this mistake and corrected

Check the use of capitals in figure, it should Figure

Response: this has been corrected throughout

Check the use of capitals in peg / PEG

Response: this has been corrected throughout

p2l10: 'Such measurements "of are" ongoing. . .'

Response: this has been changes as per the comment of reviewer 1.

p3l9: maybe: "(> 6 month)"

Response: agreed

p4l26: too many "averages" here

Response: agreed, the sentence has been rewritten as - *"The average (mean) of the maximum 3 values in the thermogram is used to extract the Tmax values reported here."*

p4l49: "mass spectrometry and CIMS" is somehow double, you may want to modify this phrase

Response: correct. Mass spectrometry is deleted from the sentence to only leave reference to CIMS

p5l18: "Tmaxes"

Response: this is the correct use here.

p7l12: the "Foley" reference is incomplete

Response: this reference has now subsequently been removed from the main text

**Anonymous Referee #3**

The manuscript, 'A method for extracting calibrated … ' describes a method of using a series of PEG compounds to calibrate the FIGAERO for determining vapor pressures of detected compounds. The work presents a useful concept that can be used by a growing number of research groups that use the FIGAERO and similar techniques to normalize (or standardize) measurements of the volatility of OA components. The manuscript is succinct, which is nice, but some potentially major details are missing. Please address/clarify the below issues that may affect the applicability of the presented concept, after which the manuscript can be considered for publication.

**Specific Comments**

A series of PEG compounds was used as calibrants to connect literature VP values of the PEG compounds to FIGAERO Tmax values. The calibration curve as defined by equation 1 is not shown on figure 4. It would appear to me that a simple exponential curve does not fit the observed values well. Please include the calibration curve on figure 4, and discuss potential reasons for the deviation from the calibration curve.

Response: this was also raised by reviewer one, please see the detailed response there.

The PEG compounds on figure 3 exhibit different desorption profiles, that is, some are much broader than others. Why?

Response: It is not clear why there are different desorption profiles seen in our data. As already noted in response to referee #1, whilst significant concentration variability and perturbed ramp rates might lead to such behaviour, our experiments were well constrained.

How were the PEG compounds introduced? Injected? Individually or together? Same heating ramp rate for each PEG? Are the four thermograms of each PEG of the same amount introduced? Have the authors tried injecting widely ranging amounts of a PEG compound?

Response: The PEG samples were manually syringed on to the filter. These were mostly completed individually but one repeat of 4 and 6 and 5 and 8 were conducted together. This is now clarified in the text.

The concentrations used are now discussed in detail as per the request of reviewer 1. The same heating ramp rate is used, the limitation of this is discussed but as a suggestion to other users it is made clear that the same ramp rate for both the calibrations and measurements should be used in order to reduce the effect of this uncertainty.

*"The same cycle was used for the single component measurements for both the GU and UMan instruments. It is however noted that the analysis provided here does not take into account the possibility of a change in ramp rate affecting the $T_{max}$. It is therefore recommended that the calibration cycles match that of the measurements."*

How much does the amount introduced affect the Tmax value? The amount of OA present can affect the Tmax values, as reported by many FIGAERO users. This issue can severely affect the applicability of the presented technique, thus the Tmax dependence on OA loading needs to be addressed carefully. Also, please consider plotting signal versus temperature, not time on figure 3.

From the concentrations used in this study there was no dependence observed in either the PEG calibrations or the chamber experiments. We do however agree that the amount of OA on the filter has the potential to affect the measured Tmax. This is now clearly discussed in the paper. As per the request of referee 1 the ranges of concentrations used for the calibrations are now included in the paper

*"As with any calibration it is recommended to use a comparable amount of calibrant material as would be expected to accumulate during the measurements, as it is noted that the amount of condensed material on the filter can affect the $T_{max}$."*

We do however reiterate that this paper is presenting this as a method to use and the applicability of the PEG series appropriate calibration of the FIGAERO. We do not suggest that equation one can be utilized widely within the community, although our data may provide a useful reference against which others with the ARI FIGAERO may compare.

The main point of the paper is to use a set of compounds that have been identified as a recommended standard for vapour pressure measurement techniques and apply them to the FIGAERO to overcome previously reported uncertainties (Bilde et al, 2015). While we agree that there are factors that can lead to the variability of the $T_{maxes}$ reported here, such as the concentrations used, we feel our method is robust.

I am particularly concerned with the concept of Tmax for compounds like PEG 4, 5, and 8, all of which show very broad desorption profiles. For these species in particular, the amount introduced, heating ramp rate, etc. can affect Tmax values greatly. Perhaps consider reporting the temperature at which half the mass comes off the filter, as opposed to Tmax.

Response: As described we have thoroughly checked all known parameters that may affect the thermograms reported here and we recognise some of the thermograms are slightly broader than expected, nonetheless as a calibration method for the FIGAERO community to use the authors feel that this is an important step to take for using the FIGAERO in this way.

As used in the Stark et al (2017) paper reporting the temperature at which half the mass comes off the filter looks to show good linearity with the $T_{max}$. Due to the much greater proportion of the community using $T_{max}$ as well as the lack of correlation with concentration and $T_{max}$ observed here, we have chosen to follow this method for the paper. The following has however been added to the paper to show this as a possible method for other FIGAERO users to employ.

*"In this study the $T_{max}$ is reported, however an alternative method, "$T_{50}$", as described in Stark et al., (2017) uses the temperature at which 50% of the signal is desorbed."*

There is brief mention of alpha-pinene oxidation, but no figures are shown and no results discussed. Please elaborate.

Response: We feel that additional information regarding these experiments is not pertinent to the paper we present here, therefore other than including the total mass measured in this study; additional information is not required in this instance.

line 26 page 4 minor manually extracted?

Response: No change has been made to this;

*"Due to the relativity small numbers of thermograms analysed from the UMan FIGAERO-CIMS, the $T_{maxes}$ from the Manchester data were manually extracted."*

line 26 page 2, not Teflon (specific to DuPont product), report specific compound like PFA or PTFE

Response: this has been specified in the text